# Changes in Prevalence and Determinants of Self-Reported Hypertension among Bangladeshi Older Adults during the COVID-19 Pandemic

**DOI:** 10.3390/ijerph192013475

**Published:** 2022-10-18

**Authors:** Sabuj Kanti Mistry, ARM Mehrab Ali, Uday Narayan Yadav, Fouzia Khanam, Md. Nazmul Huda, David Lim, ABM Alauddin Chowdhury, Haribondhu Sarma

**Affiliations:** 1ARCED Foundation, 13/1 Pallabi, Mirpur-12, Dhaka 1216, Bangladesh; 2Centre for Primary Health Care and Equity, University of New South Wales, Sydney, NSW 2052, Australia; 3BRAC James P Grant School of Public Health, BRAC University, Dhaka 1213, Bangladesh; 4Department of Public Health, Daffodil International University, Dhaka 1207, Bangladesh; 5National Centre for Epidemiology and Population Health, Australian National University, Canberra, ACT 0200, Australia; 6Department of Public Health, North South University, Dhaka 1229, Bangladesh; 7Translational Health Research Institute, School of Medicine, Western Sydney University, Campbeltown, NSW 2560, Australia; 8School of Health Sciences, Western Sydney University, Campbelltown, NSW 2560, Australia

**Keywords:** hypertension, older adults, COVID-19, determinants, prevalence, Bangladesh

## Abstract

The present study aimed to assess the changes in the prevalence and determinants of self-reported hypertension among older adults during the COVID-19 pandemic in Bangladesh. This repeated cross-sectional study was conducted on two successive occasions (October 2020 and September 2021), overlapping the first and second waves of the COVID-19 pandemic in Bangladesh. The survey was conducted through telephone interviews among Bangladeshi older adults aged 60 years and above. The prevalence of hypertension was measured by asking a question about whether a doctor or health professional told the participants that they have hypertension or high blood pressure and/or whether they are currently using medication to control it. We also collected information on the socio-economic characteristics of the participants, their cognitive ability, and their COVID-19-related attributes. A total of 2077 older adults with a mean age of 66.7 ± 6.4 years participated in the study. The samples were randomly selected on two successive occasions from a pre-established registry developed by the ARCED Foundation. Thus, the sample in the 2021-survey (round two; *n* = 1045) was not the same as that in the 2020-survey (round one; *n* = 1031) but both were drawn from the same population. The findings revealed that the prevalence of hypertension significantly increased across the two periods (43.7% versus 56.3%; *p* = 0.006). The odds of hypertension were 1.34 times more likely in round two than in the round one cohort (AOR 1.34, 95% CI 1.06–1.70). We also found that having formal schooling, poorer memory or concentration, and having had received COVID-19 information were all associated with an increased risk of hypertension in both rounds (*p* < 0.05). The findings of the present study suggest providing immediate support to ensure proper screening, control, and treatment of hypertension among older adults in Bangladesh.

## 1. Introduction

The world has been crippled by the pandemic outbreak caused by severe acute respiratory syndrome coronavirus 2 (SARS-CoV-2) since December 2019 [1]. As of 4 August 2022, COVID-19 has affected over 578 million people worldwide in at least 166 countries [2]. The number of deaths has already exceeded 6.4 million globally and is expected to increase further as the disease spreads rapidly [2]. Recent studies have shown that COVID-19 patients of older age and with a co-morbidity, such as cardiovascular disease, diabetes, hypertension, or chronic lung disease, are at a higher risk of infection with a higher mortality rate than the general population [3,4,5]. Hypertension was the most commonly reported co-morbidity in COVID-19 patients [6]. 

In recent years, hypertension, a non-communicable disease (NCD), has been recognised as the leading health issue as it affects 1.39 billion individuals worldwide [7], and the prevalence of hypertension increases with age (affecting approximately 70% of older adults) [8]. Evidence also suggests that the majority of the population with hypertension resides in low-income and middle-income countries (LMICs) [9]. Hypertension, responsible for 13% of global deaths, has become one of the most challenging concerns for world public health [10]. By 2025, hypertension-related deaths are anticipated to increase up to 30% worldwide and play a pivotal role in the rising global burden of disease and disability [11,12,13]. Experimental studies identified that SARS-CoV-2 infection could directly or indirectly trigger cardiovascular complications such as myocarditis, cardiomyopathy, and congestive heart failure [14,15]. Furthermore, there is evidence that ‘lockdown’, imposed due to COVID-19, may have exerted some negative consequences on health behaviours, such as increased smoking, poor diet, sedentary lifestyle, and increased alcohol use, resulting in an increased prevalence of chronic diseases including hypertension [16,17]. It is predicted that the COVID-19 pandemic will create another pandemic of NCDs, especially hypertension [18].

The COVID-19 pandemic has not ended yet in Bangladesh. The first wave of COVID-19 started in Bangladesh in early March 2020 and was harsh, and a second wave started at the end of March 2021 [19,20]. Despite government efforts, COVID-19 has reached all 64 administrative districts in Bangladesh, with more than 2 million confirmed cases and 29,298 deaths as of 4 August 2022 [2], with the highest death rate (45%) in the older population [21]. Notably, the aging population is also rapidly growing in Bangladesh; 13 million people are aged 60 years or above, which comprises 8% of the total population [22]. Pre-pandemic data suggests that the overall prevalence of hypertension was 31% in Bangladesh, and the prevalence was higher among the older population aged 60 years and above [23]. The uncertainties and mitigating measures related to the pandemic have changed peoples’ everyday lifestyles, and home confinement may increase the risk of hypertension Previous research also documented that aging is often associated with a lack of memory/cognitive abilities [24], which can be further aggravated during the COVID-19 pandemic [25]. The simultaneous presence of cognitive dysfunction and a high level of hypertension are reported among older adults [26,27]. Therefore, since the age of any individual is a risk factor for COVID-19 infection, cognition, and hypertension occurrence, the ongoing pandemic could significantly exacerbate the condition of hypertension among the older population in Bangladesh. 

Pre-pandemic data suggests that the prevalence of hypertension was high among the older population in Bangladesh. According to the 2011 Bangladesh Demographic and Health Survey, the prevalence of hypertension was 35% and 40% among older adults aged 60–69 years and 70+ years, respectively [28]. A recent study, conducted immediately before the pandemic, documented that nearly half of the older adults were hypertensive [29]. Previous studies also identified several factors such as age, nutritional status, and sedentary lifestyles as well as pre-existing co-morbidities associated with hypertension among the adult population in Bangladesh [30,31,32]. However, no studies have documented the prevalence of hypertension in the older population and the changes in factors associated with hypertension over time during the COVID-19 pandemic in Bangladesh. To fill this important gap, we aimed to assess changes in the prevalence of self-reported hypertension in the older population in two timeframes during the COVID-19 pandemic in Bangladesh. We also explored the changes in factors associated with hypertension in the older population during the pandemic in Bangladesh. This research is particularly important from a policy perspective, as a scientific understanding of hypertension prevalence and its determinants is critical to appropriately identify the entry points for interventions to prevent and control hypertension among Bangladeshi older adults during the COVID-19 pandemic.

## 2. Materials and Methods

### 2.1. Study Design and Participants

This repeated cross-sectional study was conducted on two successive occasions, i.e., October 2020 and September 2021, overlapping with the first and second waves of the COVID-19 pandemic in Bangladesh. Data was collected from older Bangladeshi adults aged 60 years and above through telephone interviews. As we could not conduct the data collection face-to-face during the COVID-19 pandemic because of the restrictions in movements, we utilised a pre-established registry as a sampling frame that was developed by the ARCED Foundation. This registry was developed by merging the households’ contact information extracted from ten different community-based studies conducted by the ARCED Foundation to explore different social issues during 2016–2020, which included households from all eight administrative divisions of Bangladesh. This registry has also been used as a sampling frame in our previous studies [33,34]. Considering a 50% prevalence with a 5% margin of error, at the 95% confidence level, 90% power of the test, and a 95% response rate, a sample size of 1096 was calculated. However, during the 2020 survey, only 1032 of the approached eligible participants responded to the study, resulting in an overall response rate of approximately 94%. During the 2021 survey, 1045 of the approached eligible participants responded to the study, resulting in an overall response rate of approximately 95%. Based on the population distribution of older adults by geography in Bangladesh, we adopted a probability proportionate to size (of the eight divisions) approach to randomly select the desired number of older adults from each division [22]. Notably, the samples were selected randomly from the registry on two successive occasions (the 2020 survey and 2021 survey). Therefore, the sample was not identical on the two occasions, while the population remained the same. In fact, we ensured that the samples were not overlapped during the two rounds of survey and if there was any incidence of the same sample of the first round selected in the second round, it was replaced by another random selection. The detailed sampling strategy is presented in Figure 1.

### 2.2. Measures

#### 2.2.1. Outcome Measure

The outcome of this study was self-reported hypertension. The participants were asked, “Have you told by a doctor or by a health professional that you have hypertension or high blood pressure?” The participants who answered ‘yes’ were also asked, “Are you currently taking any medicine for your hypertension?” [35]. Therefore, self-reported hypertension was defined as having been diagnosed with hypertension/high blood pressure and/or currently taking any medication for it.

#### 2.2.2. Explanatory Variables

An extensive review of available studies guided the selection of explanatory variables [36,37,38,39,40,41]. Explanatory variables considered in this study were administrative division (Barishal, Chattogram, Dhaka, Mymensingh, Khulna, Rajshahi, Rangpur, and Sylhet), age (categorized as 60–69, and ≥70), sex (male/female), marital status (married/without partner), formal schooling (without formal schooling/with formal schooling), family size (≤4 or >4), family monthly income (BDT) (<5000, 5000–10,000, or >10,000), residence (urban/rural), current occupation (employed/unemployed or retired), living arrangement (living alone or with family), walking distance to the nearest health centre (<30 min/≥30 min), memory or concentration problems (no problem/low memory or concentration), and having received COVID-19-related information from health workers (yes/no).

To explore if the participants had any memory or concentration problem, we asked the participants: “Do you have any memory (remembering things properly) or concentration (could concentrate properly while doing any action) problem?” Meanwhile, participants were considered to have received COVID-19-related information from health workers if they mentioned that they received information about the risk factors and prevention of COVID-19.

### 2.3. Data Collection Tools and Techniques

A pre-tested semi-structured questionnaire was used to collect the information via a telephone interview. Data collection was accomplished electronically using the SurveyCTO mobile app (https://www.surveycto.com/, accessed on 1 May 2022) by trained research assistants, recruited based on previous experience of administering health surveys on the electronic platform. The research assistants were trained extensively before the data collection through Zoom meetings.

The English version of the questionnaire was first translated to the Bangla language and then back translated to English to ensure the content’s consistency. The questionnaire was then piloted among a small sample (*n* = 10) of older adults to refine the language in the final version. The tool used in the pilot study did not receive any corrections/suggestions from the participants about the contents developed in the Bengali language. We found the questionnaire was reliable with a good internal consistency (Cronbach’s α, 0.72)

### 2.4. Statistical Analysis

The distribution of the variables was assessed through descriptive analyses. Given our variables’ categorical nature, Chi-squared tests were performed to compare differences in the prevalence of hypertension by explanatory variables, with a 5% level of significance. We used a binary logistic regression model to determine the changes in the prevalence of hypertension after adjusting for all potential covariates in the pooled data. We also assessed the factors associated with hypertension in the first and second rounds of the survey. We executed two separate regression models (round one and round two), and the final models were selected based on the lowest AIC (Akaike information criterion) values. The variables with *p* < 0.25 in the unadjusted analysis were only included in the multiple regression model [42]. Crude odds ratio (cOR), adjusted odds ratio (aOR), and an associated 95% confidence interval (95% CI) are reported. All analyses were performed using the statistical software package Stata (Version 14.0). 

### 2.5. Ethical Approval

This research was approved by the institutional review board of Institute of Health Economics, University of Dhaka, Bangladesh (Ref: IHE/2020/1037). We sought verbal informed consent from the participants before the survey and their participation was voluntary and without any compensation.

## 3. Results

### 3.1. Characteristics of the Participants

Table 1 shows the characteristics of the study participants by survey year. In terms of survey participant coverage, there was a significant difference across geographic areas; for example, the highest coverage was from the Dhaka division in the 2020 survey, while the highest coverage was from the Khulna division in the 2021 survey. In both surveys, most participants were 60–69 years old, male, married, without formal schooling, unemployed/retired, lived with family, and lived in rural areas (Table 1). However, participants’ characteristics (e.g., sex, marital status, education, and income) were significantly different across the survey years. Compared to the 2020 survey, a considerably lower proportion of participants in the 2021 survey was males (59% vs. 66%), married (77% vs. 81%), and without formal education (52% vs. 58%). The proportion of participants living with family, from rural areas, close to health facilities, and who had problems with memory and concentration increased significantly between the survey years. However, we found that fewer people received COVID-19-related information from health workers in the 2021 survey (Table 1).

### 3.2. Changes in the Prevalence of Hypertension

Table 2 shows the changes in the prevalence of hypertension over time and their variation and association with participants’ characteristics. As seen in Table 2, the prevalence of hypertension was significantly increased between the two survey years among the participants (43.7% versus 56.3%; *p* = 0.006). Moreover, after adjusting for all potential covariates, compared to round one, the odds of hypertension were significantly higher in round two (AOR 1.34, 95% CI 1.06–1.70) (Table 3). The final regression model was adjusted for all the covariates presented in Table 1.

We can also see in Table 2, the prevalence of hypertension was significantly increased among the participants residing in divisions of Chattogram, Rangpur, and Sylhet (43.6% vs. 56.4%; 33.3% vs. 66.7%; and 36.4% vs. 63.6%, respectively), aged 60–69 years (44.8% vs. 55.2%), females (37.7% vs. 62.3%), without partners (35.4% vs. 64.6%), residents of rural areas (40.9% vs. 59.2%), and living with a family (42.4% vs. 57.7%). A significant increase in the prevalence of hypertension was also documented among those who received no formal schooling (35.4% vs. 64.6%), had a family income of 5000–10,000 BDT (32.0% vs. 68.0%) or >10,000 BDT (49.2% vs. 50.8%), were residing with a family with a size of no more than four members (35.8% vs. 64.2%), unemployed or retired (40.1% vs. 59.9%), living near a health facility (36.5% vs. 63.5%), had memory or concentration problems (32.9% vs. 67.1%), and did not receive COVID-19-related information from health workers (38.8% vs. 61.2%).

### 3.3. Changes in Factors Associated with Hypertension

Table 4 presents the factors associated with hypertension in the first and second survey rounds. In the adjusted analysis, we found that having formal schooling, low memory or concentration, and receiving COVID-19 information were significantly associated with an increased risk of hypertension (*p* < 0.05) in both rounds. In contrast, in round 1, only participants living alone had a higher risk of hypertension than those living with a family.

## 4. Discussion

This study found that the hypertension prevalence in Bangladeshi older people increased significantly from 43.7% in 2020 to 56.3% in 2021. The prevalence of hypertension increased more among female participants than in their male counterparts. Having formal education, being unemployed or in retirement, having a low ability to concentrate, and interacting with health workers to get COVID-19-related information were also significantly associated with a higher prevalence of hypertension. To the best of our knowledge, these findings are novel in Bangladesh, meaning that no studies explored the changes in the prevalence of hypertension in older adults during the COVID-19 pandemic in Bangladesh. However, studies in other settings documented an increased prevalence of uncontrolled blood pressure among different population groups during the COVID-19 pandemic, compared to the pre-pandemic period [43,44,45,46]. For example, a longitudinal study conducted in the United States reported that the prevalence of hypertension increased significantly among female participants during the first year of the COVID-19 pandemic, from 38% in 2020 to 62% in 2021. The pandemic situation also severely affected healthcare professionals in low-income settings. A study conducted in India found that the COVID-19 pandemic significantly impacted the blood pressure of hospital staff, including doctors and nurses, with an alarming increase in risk to their health [47]. 

The COVID-19 pandemic forced many countries, including Bangladesh, to impose restrictions on the movement of citizens, including limiting social interactions, implementing lockdowns, and imposing strict quarantine protocols to prevent the spread of the virus [21,48]. These restrictions greatly impacted people’s lives and their psychosocial well-being. Frequent lockdowns and quarantines limited access to physical activity and regular exercise while increasing smoking and other unhealthy behaviours. Such harmful lifestyle choices directly affect uncontrolled blood pressure and increase the prevalence of hypertension at the population level [49]. The COVID-19 pandemic also affected access to regular health services for chronic disease management, particularly in developing countries where digitalised healthcare is not yet established. As a result, citizens could not access regular health check-ups and doctor consultations, and many healthcare facilities restricted entry or shut down due to the pandemic [50,51]. A recent study in Bangladesh also found that chronic disease management and routine health services for the older population were seriously hampered during the COVID-19 pandemic [51]. All of these factors might have contributed to an increase in the prevalence of hypertension during this pandemic. 

This study identified educational status as an important predictor of hypertension. This is because educational status is associated with health literacy, which affects the capability of an individual to attain, process, and understand the health information and services necessary to make appropriate personal decisions [52]. Education can also lead to accurate health beliefs and decrease misconceptions, thus facilitating the making of informed lifestyle choices and promoting self-advocacy. A previous systematic review found that people with low educational status were twice as likely to be hypertensive (OR 2.02, 95% CI 1.55–2.63) compared to those with a higher educational status [53]. However, the results of our analysis contradicted this evidence, as we found that participants with formal education were more likely to have hypertension than those with no formal education. This may be because the people with formal education relied more on COVID-19 information that they collected from various sources, including the internet, which might have resulted in increased fear and anxiety. Evidence suggests that increased fear and anxiety are connected to high blood pressure [54,55]. A recent systematic review of 18 studies conducted in Southeast Asia reported mixed associations between education and chronic diseases related to multimorbidity [56]. Considering such fluctuating evidence, further studies are required to better understand the association between educational status and the prevalence of hypertension. Such studies could aid in the development of appropriate interventions to control hypertension in populations with low socio-economic status. 

This study found that the unemployed or retired respondents who participated in round two of our survey study were 1.7 times more at risk of being hypertensive than the employed respondents. This may be because of the increased anxiety level among older people due to their lack of financial security caused by the pandemic. Previous studies in Bangladesh found that more than one-fourth of the participants were not receiving salaries due to the pandemic. Thus, they were more likely to suffer from anxiety and depressive symptoms [39,57]. Another important finding of our study was that participants with memory and concentration problems or low cognitive function were more likely to suffer from hypertension than those without these issues. Previous research also documented that hypertension is associated with cognitive impairment, such as loss of cognitive functions, memory, concentration, and language, which significantly decreases the quality of life with increasing age [58,59,60,61]. Previous research reported the simultaneous presence of increased hypertension and loss of cognitive abilities [26,27]. In particular, it is well established that hypertension can affect brain structure and function and it is most consistently linked to late-life cognitive decline and dementia [62]. However, whether cognitive decline increases the risk of hypertension is not well described in the literature [63]. 

Community health workers in Bangladesh have been well known for delivering health promotion activities including health education through a community-based approach [64]. Interacting with community healthcare workers may benefit patients’ health by facilitating the latter’s access to critical health information and providing psychosocial support [65]. However, this study found that participants who responded to both surveys and received COVID-19-related information from healthcare workers were more likely to suffer from hypertension than participants who did not receive COVID-19 information from healthcare workers. Although surprising, this finding provides critical information about the quality of health education-related services provided by healthcare workers during this time. Thus, healthcare workers may have lacked the COVID-19-related training needed to impart information to older people in Bangladesh. During the initial years of the COVID-19 pandemic, a vast amount of inappropriate or incorrect information was circulated across social media platforms, which may have negatively influenced healthcare workers’ knowledge. A recent WHO study found that the spread of COVID-19-related misinformation via social media and other digital platforms is a threat to global public health [66]. Therefore, it is critical to design and implement appropriate training for healthcare workers to provide correct COVID-19-related information to the older population of Bangladesh. 

### Strengths and Limitations of the Study

This study has several strengths. Firstly, this is the first study reporting the change in the prevalence of and factors associated with hypertension among the older population during the COVID-19 pandemic in Bangladesh. Second, this study represents one of the most unresearched population in Bangladesh during the COVID-19 pandemic. 

Nevertheless, the study was subjected to several limitations as well. First, we could not develop a panel; instead, we followed a repeated cross-sectional design. Thus, the change suggests a trend observed in a population and does not indicate changes experienced at the individual level. Second, our study was limited to quantitative analysis and points to potential factors. A qualitative study may shed better light on the underlying reasons for the change. In addition, the hypertension was self-reported and was therefore subjected to reporting bias. Moreover, this study did not measure cognitive function with an appropriate clinically validated tool; thus, the findings must be interpreted cautiously. 

## 5. Conclusions

The present study documented an increased prevalence of hypertension among the older population during the pandemic in Bangladesh. The study also highlighted those specific subgroups of the population, such as those having formal schooling, who had low memory or concentration, and who received COVID-19-related information, had a higher risk of increased prevalence of hypertension in both rounds. The findings are particularly crucial for policymakers and public health practitioners to emphasise the need to provide immediate support to ensure the prevention and early management of hypertension, specifically among the most vulnerable segment of the community during this pandemic. Adequate training and the involvement of community health workers could be of value in this regard. Specifically, they could undertake the tasks such as screening for hypertension and providing psychosocial supports during this emergency period. 

## Figures and Tables

**Figure 1 ijerph-19-13475-f001:**
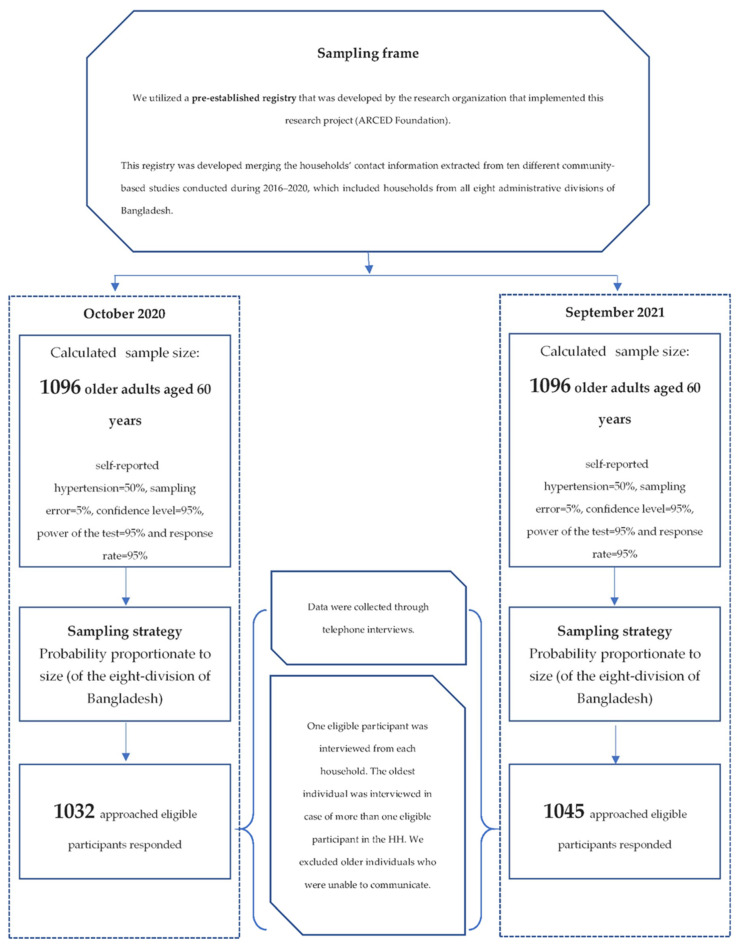
Sampling strategy and participants selection.

**Table 1 ijerph-19-13475-t001:** Characteristics of the participants (*n* = 2077).

Characteristics	Round 1		Round 2		*p* ^2^
	*n*	%	*n*	%	
Overall	1032	100.0	1045	100.0	
Administrative division					
Barishal	149	14.4	146	14.0	0.001
Chattogram	137	13.3	98	9.4	
Dhaka	210	20.4	172	16.5	
Mymensingh	63	6.1	69	6.6	
Khulna	158	15.3	198	19.0	
Rajshahi	103	10.0	145	13.9	
Rangpur	144	14.0	161	15.4	
Sylhet	68	6.6	56	5.4	
Age (year)					
60–69	803	77.8	790	75.6	0.385
>= 70	229	22.2	255	24.4	
Sex					
Male	676	65.5	620	59.3	0.004
Female	356	34.5	425	40.7	
Marital status					
Married	840	81.4	799	76.5	0.006
Without partner	192	18.6	246	23.5	
Formal schooling					
No formal schooling	602	58.3	540	51.7	0.002
Having formal schooling	430	41.7	505	48.3	
Family size					
≤4	318	30.8	347	33.2	0.243
>4	714	69.2	698	66.8	
Family monthly income (BDT) ^1^					
<5000	145	14.1	121	11.6	<0.001
5000–10,000	331	32.1	469	44.9	
>10,000	556	53.9	455	43.5	
Residence					
Urban	269	26.1	182	17.4	<0.001
Rural	763	73.9	863	82.6	
Current occupation					
Employed	419	40.6	407	39.0	0.441
Unemployed/retired	613	59.4	638	61.1	
Living arrangement					
Living with family	953	92.3	992	94.9	0.016
Living alone	79	7.7	53	5.1	
Walking distance to the nearest health centre					
<30 min	503	48.7	581	55.6	0.002
≥30 min	529	51.3	464	44.4	
Problem in memory or concentration					
No problem	782	75.8	676	64.7	<0.001
Low memory or concentration	250	24.2	369	35.3	
Receiving COVID-19-related information from health workers					
No	936	90.7	981	93.9	0.007
Yes	96	9.3	64	6.1	

^1^ BDT is approximately 0.012 USD ^2^ Chi-squared tests generated the *p* values which denotes the difference between round 1 and round 2.

**Table 2 ijerph-19-13475-t002:** Change in hypertension prevalence over the time (*n* = 2077).

Characteristics	Round 1		Round 2		*p* ^1^
	*n*	%Hypertensive	*n*	%Hypertensive	
Overall	1032	43.7	1045	56.3	0.006
Division					
Barishal	149	43.3	146	56.7	0.213
Chattogram	137	43.6	98	56.4	0.041
Dhaka	210	58.3	172	41.7	0.444
Mymensingh	63	36.8	69	63.2	0.305
Khulna	158	43.2	198	56.8	0.825
Rajshahi	103	34.8	145	65.2	0.303
Rangpur	144	33.3	161	66.7	0.024
Sylhet	68	36.4	56	63.6	0.013
Age (year)					
60–69	803	44.8	790	55.2	0.036
≥70	229	41.2	255	58.8	0.102
Sex					
Male	676	48.3	620	51.7	0.189
Female	356	37.7	425	62.3	0.014
Marital status					
Married	840	46.8	799	53.3	0.080
Without partners	192	35.4	246	64.6	0.036
Formal schooling					
No formal schooling	602	35.4	285	64.6	0.036
Having formal schooling	430	42.1	208	57.9	0.165
Family size					
≤4	318	35.8	347	64.2	0.006
>4	286	46.5	698	53.5	0.101
Family monthly income (BDT) ^1^					
<5000	145	45.8	86	54.2	0.182
5000–10,000	125	32.0	235	68.0	0.021
>10,000	202	49.2	172	50.8	0.035
Residence					
Urban	269	53.8	182	46.2	0.194
Rural	763	40.9	863	59.2	0.014
Current occupation					
Employed	419	51.9	407	48.1	0.764
Unemployed/retired	613	40.1	638	59.9	0.000
Living arrangement					
Living with family	387	42.4	392	57.7	0.003
Living alone	29	62.1	24	37.9	0.782
Distance from the nearest health centre					
<30 min	503	36.5	581	63.5	0.001
≥30 min	529	52.7	464	47.3	0.852
Problem in memory or concentration					
No problem	782	50.8	676	49.2	0.317
Low memory or concentration	250	32.9	369	67.1	0.022
Receiving COVID-19-related information from health workers					
No	936	38.8	981	61.2	<0.001
Yes	96	67.1	64	32.9	0.092

^1^ Chi-squared tests generated the *p* values which denotes the difference between round 1 and round 2.

**Table 3 ijerph-19-13475-t003:** Odds of change in prevalence of hypertension over the time (*n* = 2077).

Characteristics	cOR ^1^	95% CI	*p* ^3^	aOR ^2^	95% CI	*p* ^4^
Self-reported hypertension						
2020 Survey	*Ref*			*Ref*		
2021 Survey	1.35	1.09–1.68	0.006	1.34	1.06–1.70	0.013

^1^ Crude Odds Ratio; ^2^ Adjusted Odds Ratio; ^3^ Changes observed in binary logistic regression model; ^4^ Changes observed in binary logistic regression model after adjusting for all covariates reported in Table 1.

**Table 4 ijerph-19-13475-t004:** Changes in factors associated with hypertension among the participants.

Characteristics	Round 1 (*n* = 1032)			Round 2 (*n* = 1045)		
	aOR ^1^	95% CI	*p*	aOR	95% CI	*p*
Age (year)						
60–69	*Ref*			*Ref*		
≥70	1.35	0.91–2.02	0.138	1.20	0.84–1.72	0.308
Sex						
Male	-			*Ref*		
Female				1.15	0.79–1.67	0.479
Marital status						
Married	*Ref*			*Ref*		
Without partners	1.31	0.83–2.07	0.239	1.34	0.92–1.97	0.131
Formal schooling						
No formal schooling	*Ref*			*Ref*		
Having formal schooling	2.05	1.42–2.95	<0.001	1.66	1.21–2.26	0.002
Family size						
≤4	*Ref*			*Ref*		
>4	1.41	0.93–2.13	0.105	1.08	0.77–1.50	0.659
Family monthly income (BDT) ^2^						
<5000	*Ref*			*Ref*		
5000–10,000	0.99	0.55–1.80	0.979	0.75	0.45–1.26	0.281
>10,000	1.30	0.74–2.29	0.357	1.32	0.80–2.19	0.279
Current occupation						
Employed	*Ref*			*Ref*		
Unemployed/retired	1.36	0.93–1.99	0.110	1.70	1.15–2.50	0.007
Living arrangement						
Living with family	*Ref*					
Living alone	2.07	1.11–3.84	0.022	-		
Distance from the nearest health centre						
<30 min	-			*Ref*		
≥30 min				0.76	0.56–1.04	0.088
Problem in memory or concentration						
No problem	*Ref*			*Ref*		
Low memory or concentration	1.72	1.17–2.53	0.006	1.91	1.39–2.62	<0.001
Receiving COVID-19-related information from health workers						
No	*Ref*			*Ref*		
Yes	6.20	3.80–10.12	0.000	2.40	1.37–4.18	0.002

^1^ Adjusted Odds Ratio; ^2^ 1 BDT is approximately 0.012 USD.

## Data Availability

The data is available upon reasonable request from the corresponding author.

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
