# Peer review of "Changes in Prevalence and Determinants of Self-Reported Hypertension among Bangladeshi Older Adults during the COVID-19 Pandemic"

_ijerph, 2022, doi:10.3390/ijerph192013475_

Round 1
Reviewer 1 Report
Dear Authors,
This manuscript aimed to assess the changes in prevalence and determinants of self-reported hypertension among older adults during the COVID-19 pandemic in Bangladesh. The study does provide some novel findings on the change prevalence of hypertension among the older population during this pandemic in Bangladesh, highlighting those specific subgroups of the population had a higher risk of increased prevalence of hypertension. The theme is applied from a health point of view.
Specific comments
- Lines 87-89: The introduction does not clearly state the problem for the research. Is this research important JUST because no studies have evaluated the prevalence of hypertension?
- Lines 138-142: regarding the translation of the questionnaire into the Bangla language, in the pilot study, intraclass correlation coefficient and Cronbach’s alpha were performed to evaluate reproducibility and internal consistency?
- Table 2: include caption – “1 BDT is approximately 0.012 USD”.
- Table 3: review the caption.
- Lines 297-306: I suggest that the strengths of the study be better described. A paragraph for the strengths of the study. Another paragraph for the limitations of the study.
- It is necessary to perform a major review the formatting of the references according to the journal's rules.
- There are many references with missing information (example: ref 1; ref 11; ref 20; ref 51...).
Author Response
Response to Reviews - Changes in prevalence and determinants of self-reported hypertension among Bangladeshi older adults during the COVID-19 pandemic
Thank you for giving us the opportunity to revise and resubmit the manuscript, and for the constructive reviewers’ comments. In the following table, we include reviewers’ comments and outline our changes in responses to each comment. We have systematically incorporated the reviewers’ comments and suggestions, which has enhanced the manuscript. We are confident that the revisions have improved the quality of our manuscript.
Reviewer 1
|
Comment/Suggestion |
Authors’ Responses |
|
This manuscript aimed to assess the changes in prevalence and determinants of self-reported hypertension among older adults during the COVID-19 pandemic in Bangladesh. The study does provide some novel findings on the change prevalence of hypertension among the older population during this pandemic in Bangladesh, highlighting those specific subgroups of the population had a higher risk of increased prevalence of hypertension. The theme is applied from a health point of view. |
Thank you for your comment. |
|
- Lines 87-89: The introduction does not clearly state the problem for the research. Is this research important JUST because no studies have evaluated the prevalence of hypertension? |
Thank you for this comment. The purpose of this study is two folds, first to fill the gaps with new scientific evidence and support national policies to identify the entry points for interventions for Bangladeshi older adults. We made revisions in the last paragraph of the Introduction. Please see page 3 lines 98-112. |
|
- Lines 138-142: regarding the translation of the questionnaire into the Bangla language, in the pilot study, intraclass correlation coefficient and Cronbach’s alpha were performed to evaluate reproducibility and internal consistency? |
We found the tool was reliable with a good internal consistency (Cronbach’s α, 0.72). We added this information in page 5, lines 185-186. |
|
- Table 2: include caption – “1 BDT is approximately 0.012 USD”. |
Added as you have suggested |
|
- Table 3: review the caption. |
Revised as suggested |
|
- Lines 297-306: I suggest that the strengths of the study be better described. A paragraph for the strengths of the study. Another paragraph for the limitations of the study. |
Thanks for your suggestion. We have worked on this subsection and provided the strengths and limitations in two different paragraphs. Please see subsection 4.1. |
|
- It is necessary to perform a major review the formatting of the references according to the journal's rules. - There are many references with missing information (example: ref 1; ref 11; ref 20; ref 51...). |
We have checked the references and corrected where necessary. |
Reviewer 2 Report
Dear Authors
Kindly please find attached the comments.

Author Response
Response to Reviews - Changes in prevalence and determinants of self-reported hypertension among Bangladeshi older adults during the COVID-19 pandemic
Thank you for giving us the opportunity to revise and resubmit the manuscript, and for the constructive reviewers’ comments. In the following table, we include reviewers’ comments and outline our changes in responses to each comment. We have systematically incorporated the reviewers’ comments and suggestions, which has enhanced the manuscript. We are confident that the revisions have improved the quality of our manuscript.
Reviewer 2
|
Comment/Suggestion |
Authors’ Responses |
|
Abstract line 29-30: Prevalence of hypertension was measured by asking a question to the participants about if a doctor or health professional told them that they have hypertension or high blood pressure. What about if patients on antiHPT medications? It is known fact that patient might confused about the “diagnosis” they have and another way to confirm it is by asking the name of medications they’re on. Even though it is a telephone interview, this can be done by getting literate household to read the participants medications if there is. Otherwise the prevalence can be underrated. |
Thank you for your response. The respondents have been asked “Have you told by a doctor or by a health professional that you have hypertension or high blood pressure?” The participants who answered ‘yes’ were also asked, “Are you currently taking any medicine for your hypertension?” Therefore, self-reported hypertension was defined as having been diagnosed with hypertension/high blood pressure and/or currently taking any medication for this. We have outlined this in Abstract. Please see page 1, lines 29-32. However, we did not ask them to read the name of the medications. As the literacy rate among older adults in Bangladeshi population is 39%, so we thought using this method would not be appropriate for majority of the population. |
|
Abstract line 31: older adults It is good to add the mean age (± SD) of the participants that reflected as older adults |
We have added this information. Please see page 1 lines 33-34. |
|
Abstract line 31-32: total of 2077 older adults participated in the study: 1032 in the 2020-survey (round-1) and 31 1045 in the 2021-survey (round-2). It is also important to mention in abstract that whether the samples in 2020 are the same samples in 2021? Or different cohort? How to ensure the cohort in 2020 was not the same cohort in 2021? It is important to outline the sampling methods of the data collection in the abstract. This is to ensure true prevalent. |
We have now clarified this in the Abstract. Pleas see page 1, line 33-37. |
|
Abstract line 34-36: We also found that having formal schooling, poorer memory or concentration, and having had received COVID-19 information to be associated with an increased risk of hypertension in both rounds (P <0.05). Please add some more info on the methodology – what are the determinants explored? e.g: sociodemo, knowledge on C19? Or cognitive/memory? |
We have added this information in the Abstract. Please see page 1, lines 32-33. |
|
Abstract 33-34: The odds of hypertension were significantly higher 33 among the participants in round 2 than those in the round 1 (AOR 1.34, 95% CI 1.06-1.70). Suggest to phrase: The odds of hypertension is 1.34 times more likely in round 2 than those in round 1 cohort (AOR 1.34, 95% CI 1.06-1.70). |
Thank you for your comment. We have revised the line accordingly. Please see page 1, lines 39-42. |
|
Abstract Conclusion: The increased prevalence of hypertension among the older adults in Bangladesh since the COVID-19 pandemic may signal the precipitation and activation of chronic conditions. What chronic conditions represent the findings here? |
We have revised the Conclusion of the Abstract. Please see page 1, lines 44-47.
|
|
Keywords Suggest to name the relevant determinants (?cognitive/ memory) What about “prevalence”? |
Thank you for your suggestion. We revised the keywords and added determinants and prevalence. |
|
Intro line 67: The COVID-19 pandemic has not ended yet in Bangladesh, the first wave was harsh Please mention when was this in Bangladesh? |
Revised and added the timeline accordingly. Please see page 2 line 77-78. |
|
Lack information on the purpose of asking concentration / memory Please add LR on this, why is it important to ask this? Related to C-19? |
We have added this information in the Introduction. Please see page 2, lines 88-92. |
|
Study design line 101-103: This registry was developed merging the households’ contact information extracted from ten different community-based studies conducted during 2016–2020, which included households from all eight administrative divisions of Bangladesh. Please cite the published community-based study mentioned and explained the justification of the sampling frame taken. What purpose of the community based-study? |
More details of the registry are now provided along with citations of the papers which employed this registry as a sampling frame. Please see page 3 lines 115-126 |
|
Study design: Please explain whether the first cohort were the same as second cohort? If you need the same cohort, how do you decide on the sampling? Convenient? Random from the list? Do you consider f/up the same patient during second round? or different cohort? If not, what were the methods to ensure no overlapping between samples? |
We have clarified this in the revised manuscript. Please see page 3 lines 131-139. |
|
Outcome measure, line 116: The participants who answered ‘yes’ were also asked, “Are you currently taking any medicine for your hypertension?” Please rephrase in abstract on the questions on medications |
The abstract is revised accordingly. Please see page 1 lines 29-32. |
|
Variable definition line 128-130: memory or concentration problems (no problem/low memory or concentration), and receiving COVID-19 related information from health workers (yes/no). What was the exact phrasing for memory/concentration? Did you use validated/reliable questions/phrasing to identify/determine low memory/concentration? Please cite the validated/reliable tools used. The information of C-19, either heard about C-19 or what kind of related information? Please outline clearly. |
More details on these two variables are provided in the revised manuscript. Please see page 5, lines 168-173. |
|
Data collection tools: The English version of the questionnaire was first translated to Bangla language and then backtranslated to English to ensure the contents’ consistency. Please outline the statistical analysis/ outcome results of the tools used. At least internal consistencies (cronhbach alpa) of the translated questions used. Ant test retest reliability done? How are you going to ensure the reliability / validity of the questions used. |
We found the tool was reliable with a good internal consistency (Cronbach’s α, 0.72). We added this information in page 5 lines 185-186. |
|
Response rate: What was the response rate? Out of the phone calls made, how many has hypertension and agreed/consented to the survey? Please make a flow chart of the data collection/sampling of samples. |
Overall response rate was 94% and 95% respectively for the two rounds of the surveys. This was reported in the method section, under study design and participants. We made a flow diagram (Figure 1) to better represent the sampling procedure. |
|
Ethical approval: Any ethical consideration obtained? |
Information on Ethical Approval is added in the revised manuscript. Please see page 6 lines 200-204. |
|
Statistical analysis line 143: If same cohort used, then you need to use repeated measure analysis if different cohort used, what are the methods used to compare between the 2 different cohort? Please explain |
We used different cohort in two timeframes. We performed Chi square tests to note the change in two timeframes in a descriptive analysis. We used a binary logistic regression model to determine the changes in the prevalence of hypertension after adjusting for all potential covariates in the pooled data. We have executed two separate regression models (round 1 and round 2) to identify the factors associated with hypertension in these two timeframes. This information is presented in Statistical analysis subsection. |
|
Table 1: What the analysis used to get the p value? Please outline at the footnote of the table. The p value is difference within the group categories? E.g: age 60-69 vs >70 or between cohort 1 and cohort 2? |
This information is added as footnote in Table 1. |
|
Table 2: What the analysis used to get the p value? Please outline at the footnote of the table |
This information is added as footnote in Table 2. |
|
Table 3: Changes in prevalence of hypertension after adjusting for potential covariates. What variables were adjusted in the models? Please outline in the footnote the adjusted potential covariates. What methods you used in the regression? |
This information is added as footnote in Table 3. |
|
Table 4: I like your table 4. Simple yet meaningful. |
Thank you for your appreciation. |
|
Discussion 275-277: However, this study did not measure the cognitive function with an appropriate clinically validated tool. The causal relationship between cognitive decline and hypertension remains unknown [56]. Therefore the current findings need to be interpreted with caution. Please look for LR the assoc between cognitive decline & HPT (I’m sure it’s a distinct information). Although your study did not use validated tools, at least the discussion can vouch on the importance of this to be looked at in the future as the LR holds. |
We have provided more details on this issue in the Discussion and highlighted not using valid tool to measure cognitive abilities of the participant as a limitation. Please see page 11, lines 328-333 and page 12, lines 364-366. |
|
Limitations line 301: we followed a repeated crosssectional design Is repeated cross-sectional design an established study design terms? Please use the terms appropriately for your study. |
Repeated cross-sectional study design has been documented as an established research method in several literature. Please refer to the following literature: Jessiman-Perreault, G., Li, A., Frenette, N., & Allen Scott, L. (2022). Investigating the early impacts of the COVID-19 pandemic on modifiable risk factors for cancer and chronic disease: a repeated cross-sectional study in Alberta, Canada. Canadian Journal of Public Health, 1-14. Williams, A. J., Wyatt, K. M., Williams, C. A., Logan, S., & Henley, W. E. (2014). A repeated cross-sectional study examining the school impact on child weight status. Preventive Medicine, 64, 103-107. Beidas, R. S., Williams, N. J., Becker-Haimes, E. M., Aarons, G. A., Barg, F. K., Evans, A. C., ... & Mandell, D. S. (2019). A repeated cross-sectional study of clinicians’ use of psychotherapy techniques during 5 years of a system-wide effort to implement evidence-based practices in Philadelphia. Implementation Science, 14(1), 1-13. Pan, X. (2022). Repeated Cross-Sectional Design. In Encyclopedia of Gerontology and Population Aging (pp. 4246-4250). Cham: Springer International Publishing. |